# Visible Light-Driven *p*-Type Semiconductor Gas Sensors Based on CaFe_2_O_4_ Nanoparticles [note 1]

**DOI:** 10.3390/s20030850

**Published:** 2020-02-05

**Authors:** Olga Casals, Andris Šutka, Tony Granz, Andreas Waag, Hutomo Suryo Wasisto, Joan Daniel Prades, Cristian Fàbrega

**Affiliations:** 1MIND-IN2UB, Department of Electronic and Biomedical Engineering, Universitat de Barcelona, 08028 Barcelona, Spain; ocasals@el.ub.edu; 2Institute of Semiconductor Technology (IHT) and Laboratory for Emerging Nanometrology (LENA), Technische Universität Braunschweig, 38106 Braunschweig, Germany; t.granz@tu-braunschweig.de (T.G.); a.waag@tu-braunschweig.de (A.W.); h.wasisto@tu-braunschweig.de (H.S.W.); 3Research Center for Physics, Indonesian Institute of Sciences (LIPI), Tangerang Selatan 15314, Indonesia; 4Research Laboratory of Functional Materials Technologies, Faculty of Materials Science and Applied Chemistry, Riga Technical University, 1048 Riga, Latvia; andris.sutka@rtu.lv

**Keywords:** calcium iron oxides, CaFe_2_O_4_, LED, metal oxides (MOx), *p*-type gas sensors, room temperature, visible light activation

## Abstract

In this work, we present conductometric gas sensors based on *p*-type calcium iron oxide (CaFe_2_O_4_) nanoparticles. CaFe_2_O_4_ is a metal oxide (MOx) with a bandgap around 1.9 eV making it a suitable candidate for visible light-activated gas sensors. Our gas sensors were tested under a reducing gas (i.e., ethanol) by illuminating them with different light-emitting diode (LED) wavelengths (i.e., 465–640 nm). Regardless of their inferior response compared to the thermally activated counterparts, the developed sensors have shown their ability to detect ethanol down to 100 ppm in a reversible way and solely with the energy provided by an LED. The highest response was reached using a blue LED (465 nm) activation. Despite some responses found even in dark conditions, it was demonstrated that upon illumination the recovery after the ethanol exposure was improved, showing that the energy provided by the LEDs is sufficient to activate the desorption process between the ethanol and the CaFe_2_O_4_ surface.

## 1. Introduction

Metal oxide semiconductors have shown the best characteristics in term of sensitivity, selectivity, and stability in gas sensor technology. The development of new materials based on different methods, techniques and working principles has been carried out to achieve the best performance. However, the operating temperatures of metal oxide gas sensors are usually above 150 °C to activate the absorption and desorption processes between the targeted gases and the surfaces of the materials [1,2,3,4,5,6,7,8]. The high operating temperature is one of the main drawbacks of metal oxide-based gas sensors because it ultimately results in high power consumption and undesirable long-term drift problems caused by sintering effects in the metal oxide grain boundaries, yielding poor selectivity and stability [9,10]. Another disadvantage of the metal oxide-based gas sensors with the high operating temperature is related to safety as they are used in hazardous environments containing explosive or flammable gas species [10]. There are plenty of methods that had been studied to reduce the power consumption of gas sensors [11,12,13,14,15,16,17]. Recently, attempts to reduce high power consumption and working temperature needs have been centered on the fabrication of more efficient heating systems, tuning of the active sensing layer (surface functionalization or use of hybrid materials), the use of piezoelectric and triboelectric effects [18,19], and the use of different energy sources (e.g., low-power ultraviolet light-emitting diode (UV-LED) light) [20,21,22,23,24,25]. Among them, light-activated sensors have shown high potential [26,27] because the possibility to drastically reduce the power consumption by scaling down the light sources (LED platforms) or the possibility to use other technologies incompatible with heat-driven sensors (i.e., functionalization) [28]. This has been supported by the fact that micro- and nano-LEDs have been continuously researched and enhanced in terms of their performance and function for sensing applications [29]. Moreover, a novel lift-off process has been recently introduced based on femtosecond laser processing [30] to obtain free-standing LED chips that can be combined with sensing active materials to develop such integrated micro-light plates [31,32].

During the last decades, although other types of sensor systems were introduced to the environmental community like micro-/nanoelectromechanical gravimetric gas and nanoparticle sensors [33,34,35,36,37,38,39,40,41,42], conductometric gas sensors were still in high demand and widely investigated due to several advantages, e.g., low cost and flexibility in production, the large number of detectable gases/possible application fields, simplicity in measurement setup, and the ease of miniaturization for portable instruments [43]. However, most of the available conductometric sensing devices are based on *n*-type materials (e.g., ZnO and SnO_2_) with wide band gap which needs UV light to activate the absorption and desorption processes occurring in the surface of the sensing material [44,45,46,47,48]. The energy and efficiency of UV LEDs are far from being competitive compared to those of visible LEDs. Therefore, materials with lower bandgaps that are suitable with the energy of visible light will be desirable for gas sensor devices operated at room temperature.

In this paper, we present a new *p*-type metal oxide used as a sensing material, which has several advantages over *n*-type counterpart [49], e.g., (1) ability to chemisorb the higher concentrations of oxygen molecules, since the formation of a hole-accumulation layer (HAL) in *p*-type oxide semiconductors is not limited by concentrations of free charge carriers [50]; (2) capability to promote selective oxidation of various volatile organic compounds (VOCs) [51,52]; and (3) lower humidity dependence [53]. CaFe_2_O_4_ nanoparticles are *p*-type metal oxides with a band gap of ~1.9 eV making it a suitable candidate for a visible light activated gas sensor. This material stands out as a potential candidate [7] and has a suitable band gap [54,55,56] for visible light-driven gas sensors at room temperature. The experiments were conducted under light irradiation with different wavelengths of visible light at room temperature. Based on the results, light illumination effects during the experiment were proposed and investigated. Applying various LED wavelengths had a significant contribution for the sensor behavior at the absorption and desorption processes.

## 2. Materials and Methods

### 2.1. Materials

CaFe_2_O_4_ nanoparticles were produced by sol-gel auto-combustion method. Iron nitrate nonahydrate (Fe(NO_3_)_3_·9H_2_O, >98%) and calcium nitrate tetrahydrate (Ca(NO_3_)_2_·4H_2_O, >99%) with the determined metal ratio of 2:1 were dissolved in Milli-Q water. Then citric acid monohydrate (C_6_H_8_O_7_·H_2_O, >99%) was added to the nitrate solution. The molar ratio of nitrates to citric acid was 1:1. For the combustion reaction, citric acid carboxylate groups and metal nitrates act as reducing and oxidizing agents, respectively. Ammonium hydroxide (NH_4_OH) solution of 26% in water was added to the nitrate–citrate solution for improving the chelation of metal cations to citrates. To avoid unwanted results and the completion of the chelation, a pH value of 7 was chosen, because citric acid is weakly dissociated at low pH. All chemicals were purchased from Sigma–Aldrich and used as received without additional purification. The obtained solution was poured into a 100-cm^3^ corundum crucible and evaporated at 80 °C under magnetic stirring. The viscous gel was obtained by evaporating water. Moreover, the gel was dried for 24 h at 60 °C to remove water residues. Then, the gel was heated to 250 °C to initiate a self-sustaining combustion reaction and produce ferrite powder. An annealing process was conducted to as-prepared powders at 850 °C for 3 h for use in the gas-sensitive active layers [7].

### 2.2. Material Characterization 

The samples were analysed using X-ray powder diffraction (XRD) recorded at 2θ from 10° to 60° at a scanning rate of 1° min^−1^ using an Ultima+ X-ray diffractometer (Rigaku, Japan) with Cu Kα radiation. An FEI Focused Ion Beam Scanning Electron Microscope (FIB-SEM) Helios Nanolab 600 equipped with Oxford Inca 350 and an X–Max 50 mm SDD-type detector were used to study morphology and elemental composition of the as-prepared powders and sintered pellets.

### 2.3. Sensor Preparation 

Gold-interdigitated electrodes (Au–IDE) on glass (MicruX Technology, Asturias, Spain) were used as an electronic platform for measuring the electrical characteristics of the CaFe_2_O_4_. The IDE size is (10 × 6 × 0.75 mm), with 90 pairs of electrodes having a pitch of 10 µm and a line width 10 µm. CaFe_2_O_4_ nanoparticles were deposited on the surface of the Au–IDEs by drop casting, followed by a spin-coating process. After setting the substrate on the sample holder of the spin coater, 5 µL of CaFe_2_O_4_ nanoparticle 10 mg/l suspensions in ethylene glycol were deposited onto Au–IDE by a single layer spin coater and spin-coated at 2000 rpm for 40 s in the air and dried on the hot plate at 90 °C for a few minutes in order to evaporate the solvent. To attain the desired thickness of CaFe_2_O_4_ nanoparticles film, the above procedure was repeated 3 times. Afterwards, an annealing process at 450 °C for 1 h with a ramping level 5 °C/min was applied to fix the material onto the substrate and achieve good electrical contact with the Au–IDEs. 

### 2.4. Sensor Measurement 

Gas-sensing experiments were conducted in a customized chamber of 200 mL in volume. The gas flow was maintained stably at 200 mL/min during all the measurements. Reference gaseous atmospheres were provided by independent mass flow controllers blending synthetic air (SA) and ethanol (100 ppm in SA). To investigate CaFe_2_O_4_ optoelectronic properties by activating visible light, resistance measurements were conducted under synthetic air flow with different LED wavelengths (i.e., blue (465 nm), green (520 nm), yellow (590 nm), and red (640 nm)) and in dark condition (without illumination). To determine the sensitivity of the CaFe_2_O_4_ sensors towards reducing gases, different concentrations of ethanol vapors from the low to the high concentration (i.e., 10, 20, 30, 50, and 100 ppm) were then applied to the chamber under LED irradiation. The response was defined as [(R_g_ − R_a_)/R_a_] × 100%, where R_a_ and R_g_ are the electrical resistances of the sensor in the air and when exposed with ethanol, respectively. The response and recovery times were defined as the times needed by a sensor to achieve 90% of the total resistance change during the adsorption and desorption process, respectively. All experiments were performed at room temperature. 

## 3. Results and Discussion

### 3.1. Sensor Characterization Results 

The sample as-prepared powder was tested using XRD as shown in Figure 1a. The mixture of various compounds (e.g., CaCO_3_ (ICDD 04-007-4989), Fe_2_O_3_ (ICDD 00-002-1047), γ-Fe_2_O_3_ (ICDD 00-004-0755), and CaFe_2_O_4_ (ICDD 04-007-4989)) was formed during the auto-combustion reaction. Nevertheless, the admixture of different compounds crystallizes to pure CaFe_2_O_4_ compound (Figure 1b) after annealing at 850 °C for 3 h. The peak positions correspond well to the CaFe_2_O_4_-type orthorhombic unit cell (ICDD 04-007-4989) with lattice constants (Pnam) of (a) 9.160 Å, (b) 10.670 Å, and (c) 3.012 Å. In this case, the observation of additional impurity phases was not conducted.

A scanning electron microscope (SEM) image of an as-prepared powder cross-section of gas sensor pellet annealed at 850 °C for 3 h shown in Figure 2a. The as-prepared sample powders are composed of amorphous–like anisotropically shaped and closely packed grains. Figure 2b shows a surface sensor on the Au-IDE after being prepared by the spin-coating process and annealed at 450 °C for 1 h. The porous structures of interconnected grains were observed, while the grains keep their anisotropic shape. The size of individual grains of smaller dimensions varies from 70 to 300 nm, while the length of anisotropic nanoparticles is up to 650 nm. Particles are very well interconnected and fused together, at the same time maintaining open structures for gas diffusion. Gas-accessible microstructures are preferred for the high gas response. Due to the particle size and form, the coating result was non-uniform on the IDEs. That was not the best result to make a uniform layer by a spin-coating process. However, knowing the response to the ethanol vapors will make a good opportunity to introduce this material as a potential candidate for *p*-type semiconductor which has a suitable bandgap for visible light-driven gas sensors at room temperature. Another possible method to make the uniform layer on the IDE is by the screen-printing process which is a widespread method in the industry of metal oxide gas sensors.

Figure 3a shows the optical adsorption UV–vis diffuse reflectance spectra (DRS) of CaFe_2_O_4_ nanoparticles and Tauc’s plot approach to determine the bandgap. It has confirmed the adsorption spectra of CaFe_2_O_4_ in the visible light range (400–700 nm). The absorption peak at 425–455 nm (dashed green line/pattern) is due to the maximum adsorption spectra of CaFe_2_O_4_ nanoparticles. The dashed red line is the linear fit absorption spectra which are leading to the optical bandgap value. Its optical bandgap was determined to be ~1.9 eV according to the energy dependence relation of ${\left(\alpha h\nu \right)}^{2}=A\left(h\nu -E_{g}\right)$, where *α* and *E_g_* are the absorption coefficient and the bandgap of CaFe_2_O_4_, respectively. In addition, it can be seen that the material shows efficient visible light absorption spectra and the bandgap of CaFe_2_O_4_. Figure 3b shows the optoelectronic properties of CaFe_2_O_4_ under LED illumination with different wavelengths and intensities of visible light without gases. The responses indicated that the visible light is suitable for this material because its energy is equal or larger than the bandgap of CaFe_2_O_4_. It confirms that the response and recovery times depend on the energy and intensity of LEDs. When the light is ON (photo-activated), electron-hole (e-h) pairs are generated in CaFe_2_O_4_ and will interact with an oxygen molecule and pre-chemisorb oxygen ion in the surface, thus facilitating their chemisorption and increasing the majority charge in CaFe_2_O_4_. This reaction will form a hole-accumulation layer, leading to decrease in electrical resistance. On the contrary, when the light is OFF the recombination process leads to an increase of the resistance to the initial value. According to the results, visible light activation works properly in this material. Moreover, the density of majority charge carriers is not only related to the intensity of visible light, but also to the absorption at this particular energy. In thin nanoparticle films, higher absorption leads to a larger generation rate of e-h pairs, and hence to a stronger impact on desorption. 

To investigate the sensitivity of the CaFe_2_O_4_ sensors towards reducing gases (i.e., ethanol), different concentrations of ethanol vapors (i.e., from 10 to 100 ppm) were then applied to the chamber under LED illumination. The first phenomenon to be noticed was that the resistance increased in the presence of ethanol (reducing gas), confirming that CaFe_2_O_4_ is a *p*-type material. For comparison, the experiments were also performed by introducing another reducing gas (i.e., NH_3_), as shown in Appendix A and the oxidizing gas (i.e., NO_2_) with different concentrations (Appendix A). Figure 4a–e shows the dynamic response to different concentrations of ethanol under blue, green, yellow and red LED illumination and also in dark condition (without illumination). The response comparison toward different conditions is shown in Figure 4f. The blue LED had better sensitivity than the others because of its energy. The higher energy and intensity are illuminated as the material increases the energetic state and density of charge carrier on the surface, which influences the sensitivity while being exposed to the target gas. The maximum response corresponding to adsorption spectra is 3.6% at 100 ppm of ethanol for blue LED. In this case, the response and recovery times were ~18 min and ~41 min, respectively. The results obtained provide evidence that CaFe_2_O_4_ is a good candidate for visible light-driven gas sensor because of its suitable bandgap (energy of visible light spectra is 1.9–2.7 eV). The sensor responses from other LEDs are too slow compared to that from blue LED, which can be due to their insufficient energy to detect gas.

Some responses were, however, also found in dark conditions. The sensitivity in dark conditions (Figure 4f) indicates that *p*-type material has the ability to chemisorb the higher concentrations of oxygen which react with gases since the formation of a hole-accumulation space charge layer is not limited by concentrations of free charge carriers [50] despite no illumination of light. The light irradiation has not only provided visible light activation on CaFe_2_O_4_, but also contributed to the desorption process when the ethanol was removed from the chamber. The energy provided by LEDs is sufficient to activate the desorption process between ethanol and the surface of CaFe_2_O_4_. In a dark condition, there was no external energy to break the bonding of target gases on the surface sensing. Thus, the sensor signal was not able to be well recovered. The maximum recovery ability in the dark condition is 35% at 100 ppm of ethanol and the average recovery ability is less than 25% for all concentrations of ethanol.

### 3.2. Sensing Mechanism 

The working principle of gas sensors based on metal oxide depends on chemisorbed oxygen molecules on the surface (i.e., adsorption and desorption), which ionize into species such as $O_{2}^{-},{\;O}^{-}$ and $O^{2-}$ by taking electrons near the surface of the metal oxides. Generally, ionosorption species of $O_{2}^{-},{\;O}^{-}$ and $O^{2-}$ are known to be dominant at <150 °C, between 150 and 400 °C, and at >400 °C, respectively [57]. In case of a *p*-type metal oxide formed by an aggregate of nanoparticles the conduction mechanism is governed by the grain boundaries. However, unlike in the case of *n*-type metal oxide materials where the outer shell of the nanoparticles is “insulating” because of the ionosorption of oxygen species, in *p*-type metal oxides the outer shell develops a hole accumulation layer (i.e., “conducting” layer). Figure 5a shows the condition of CaFe_2_O_4_ when it is exposed to air in the dark, in which the adsorbed oxygen molecules trap electrons from the valence band of CaFe_2_O_4_ and form pre-chemisorbed oxygen ion $\left(O_{2}^{-}\right)$ on the surface at room temperature [57,58]. Pre-chemisorbed $O_{2}^{-}$ on the surface results in the presence of a high-conductivity hole-accumulation region in the surface layer of CaFe_2_O_4_. Consequently, the energy bands bend upward near the surface of CaFe_2_O_4_ (Figure 5d) in comparison with the flat band situation before any surface reaction (Figure 5c) [59]. In the dark condition, the pre-chemisorbed oxygen ion is thermally stable and difficult to remove from the surface of CaFe_2_O_4_ at room temperature due to the large absorption energy [60]. The kinetic reaction can be explained as follows [6]: (1)$$O_{2\;\left(g\right)}↔O_{2\left(\mathrm{ads}\right)}$$
(2)$$O_{2\left(\mathrm{ads}\right)}+e^{-}↔O_{2\left(\mathrm{ads}\right)}^{-}$$

When the light illuminates the materials (Figure 5b), electrons are excited from the valence band to conduction band and electron-hole pairs are generated. The holes react with the pre–chemisorbed $O_{2\left(\mathrm{ads}\right)}^{-}$ to form oxygen molecules which will be desorbed from the surface of CaFe_2_O_4_
$\left(h^{+}+O_{2\left(\mathrm{ads}\right)}^{-}↔O_{2\left(g\right)}\right)$. At the same time, new oxygen molecules will be adsorbed and capture the photo-electrons to form photo-induced oxygen ions: $\left(O_{2\left(g\right)}+e^{-}\left(h\nu \right)↔O_{2\left(\mathrm{ads}\right)}^{-}\left(h\nu \right)\right)$. The net result of these adsorbtion and desorbtion processes of oxygen molecues is that the photoinduced holes acumulate into the surface increasing the width of the hole-accumulation layer (Figure 5e). Consequently, the resistance of CaFe_2_O_4_ decreases in this reaction. The reaction can be explained as in the following equation:(3)$$\left(h^{+}+e^{-}\right)\left(h\nu \right)\to \{\begin{array}{c} h^{+}\left(h\nu \right)+O_{2\left(\mathrm{ads}\right)}^{-}↔O_{2\left(g\right)}\\ O_{2\left(g\right)}+e^{-}\left(h\nu \right)↔O_{2\left(\mathrm{ads}\right)}^{-}\left(h\nu \right) \end{array}$$

Figure 6 shows a scheme of the sensing mechanism when the material is exposed to the target gas (ethanol vapors) under illumination together with the dynamic response of the sensor (green line) and the corresponding energy band diagram for each situation. When the sensor is exposed to the ethanol vapors, the ethanol molecules are absorbed on the surface and react with photo-induced oxygen ions to form water vapor (H_2_O) and CO_2_ consuming photo-induced oxygen ions from the surface by releasing electrons (Figure 6b). The reaction can be described as follows:(4)$$O_{2\left(g\right)}+e^{-}\left(h\nu \right)↔O_{2\left(\mathrm{ads}\right)}^{-}\left(h\nu \right)$$
(5)$$2C_{2}H_{5}{\mathrm{OH}}_{\left(\mathrm{ads}\right)}+O_{2\left(\mathrm{ads}\right)}^{-}\left(h\nu \right)↔2{\mathrm{CH}}_{3}{\mathrm{COH}}_{\left(\mathrm{ads}\right)}+2H_{2}O+e^{-}$$
(6)$${\mathrm{CH}}_{3}{\mathrm{CHO}}_{\left(\mathrm{ads}\right)}+5O_{\left(\mathrm{ads}\right)}^{-}\left(h\nu \right)↔2{\mathrm{CO}}_{2}+2H_{2}O+5e^{-}$$

The released electrons will return to the valence band and cause a decreasing concentration of oxygen ions in the surface, resulting in electron-hole compensation and eventually narrowing the hole–accumulation layer. This narrowing process results in an increased resistance when a reducing gas is introduced [50], as shown in Figure 6e. When ethanol vapors are removed from the chamber, the remaining ethanol molecules adsorbed in the surface of the material will eventually desorb through reactions (5) and (6) and be replaced again by adsorbed oxygen molecules returning to the original situation (increase the concentration of hole and the width of the HAL and also resulting in a decrease the electrical resistance of CaFe_2_O_4_ (Figure 6c,f). However, the energy needed to either desorb ethanol from the surface or induce reactions (5) and (6) could be higher than that provided by the incident photons, and therefore some of the adsorbed ethanol molecules (or acetaldehyde from reaction (5)) will remain attached to the surface resulting in a state slightly different from the original with a different resistance.

## 4. Conclusions

CaFe_2_O_4_ nanoparticles have been synthesized by a sol-gel auto-combustion method resulting in unconventional metal oxides with bandgap of around 1.9 eV, which is suitable for visible light spectra. Light-activated room-temperature gas sensors based on this material has been tested and validated. The maximum responses toward ethanol vapor and recovery time were 3.6% at 100 ppm, ~18 min and ~41 min, respectively. The maximum response corresponds to the maximum absorption spectra (425–455 nm) of CaFe_2_O_4_ based on results of optical absorption UV–vis diffuse reflectance spectra. The energy provided by the LEDs is sufficient to activate the desorption process between the ethanol and the surface of CaFe_2_O_4_. In addition, it has been confirmed that visible light activation contributed to breaking the bonding of target gases from the surface sensing.

## Figures and Tables

**Figure 1 sensors-20-00850-f001:**
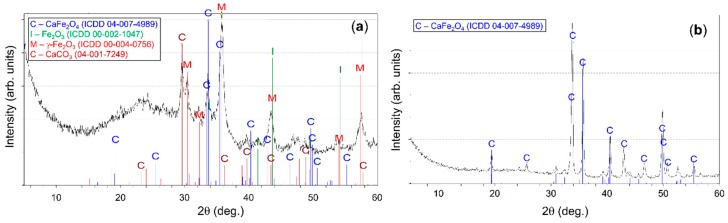
(**a**) X-ray diffraction (XRD) pattern of as–prepared and (**b**) annealed powders. Reproduced with permission from [7].

**Figure 2 sensors-20-00850-f002:**
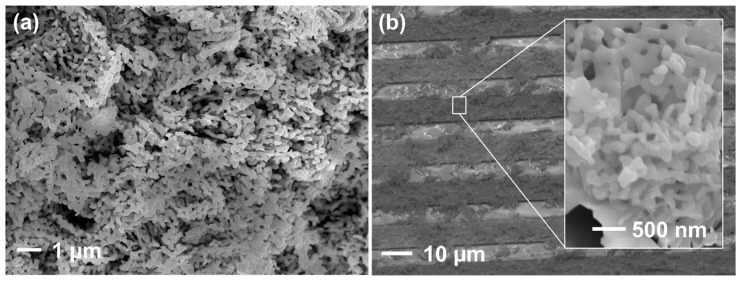
(**a**) Scanning electron microscope (SEM) image of as-prepared powder cross-section of gas sensor pellet annealed at 850 °C for 3 h and (**b**) CaFe_2_O_4_ nanoparticles on the gold-interdigitated electrodes (Au–IDE) annealed at 450 °C for 1 h prepared by the spin coating process.

**Figure 3 sensors-20-00850-f003:**
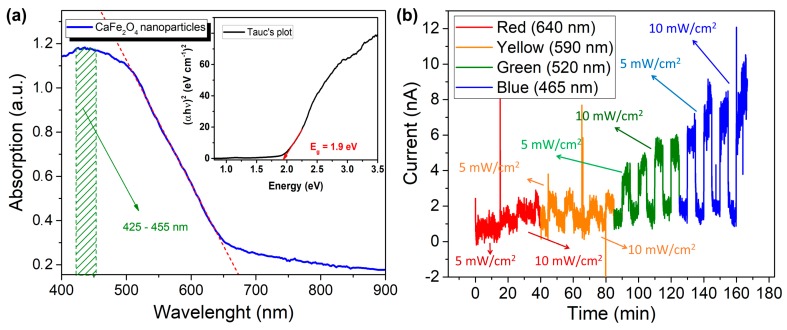
Optoelectronic characteristics of CaFe_2_O_4_ nanoparticles. (**a**) Ultraviolet–visible (UV–vis) diffuse reflectance spectra (DRS) and Tauc’s plot (**b**) CaFe_2_O_4_ behavior with different wavelengths and intensities (i.e., 5 and 10 mW/cm^2^) of light-emitting diode (LED) confirming that CaFe_2_O_4_ is suitable with the visible light energy.

**Figure 4 sensors-20-00850-f004:**
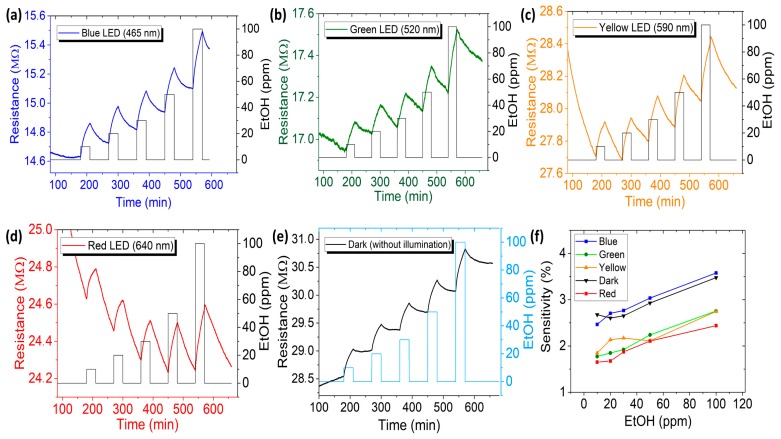
Dynamic responses of CaFe_2_O_4_ in varied vapor concentrations (i.e., 10, 20, 30, 50, and 100 ppm) of ethanol under activation of (**a**) blue (465 nm), (**b**) green (520 nm), (**c**) yellow (590 nm), (**d**) red (640 nm) LEDs and (**e**) dark condition (without illumination). (**f**) Comparison of the sensor sensitivity under visible light exposures and dark condition.

**Figure 5 sensors-20-00850-f005:**
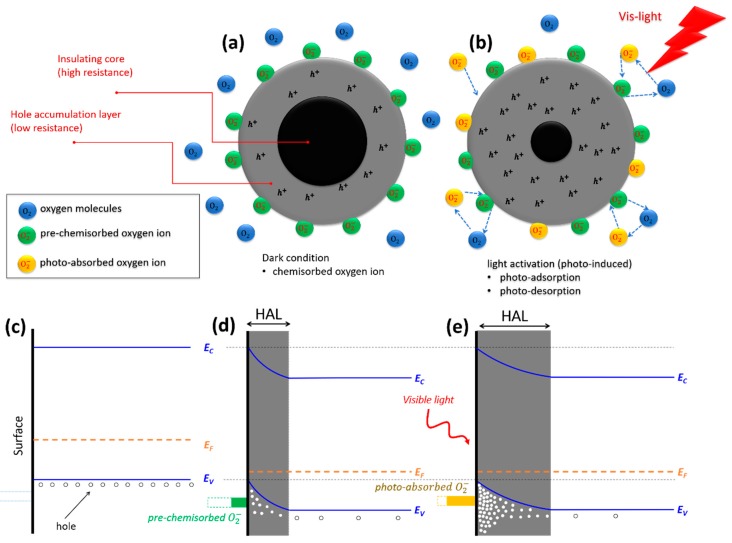
Reaction of pre-chemisorbed oxygen and sensing mechanism of *p*-type CaFe_2_O_4_ nanoparticles based on density of charge in the core shell and energy band diagram (**a**) dark condition (**b**) photo-activation (i.e., light is ON) (**c**) condition prior to any surface reaction (**d**) trapping electron from the valence band due to the pre-chemisorption process in dark condition results in the formation of a hole-accumulation layer (HAL) (**e**) photo-activation process increases the density of majority charge (hole) resulting in a widening of HAL.

**Figure 6 sensors-20-00850-f006:**
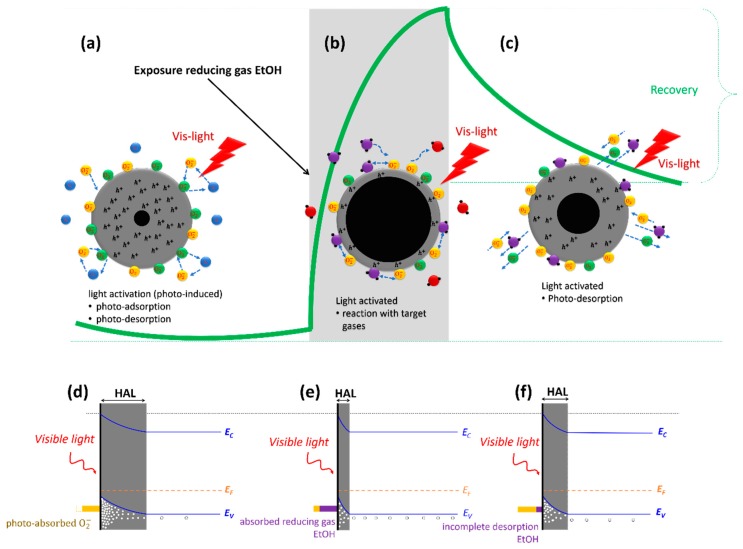
The dynamic response (the green line) toward reducing gas (i.e., ethanol vapors) under visible light irradiation that corresponded to: (**a**) photo-activation before target gas exposure has been introduced, (**b**) target gas interacts with photo-induced oxygen ion and attaches on the surface (photo-adsorption process), (**c**) photo-desorption process when target gas is detached from the surface, (**d**) energy band diagram photo-activation before interaction with target gas, (**e**) photo-adsorption process causes a decreasing the width of HAL and increasing of the electrical resistance, and (**f**) photo-desorption process causes an increasing the width of HAL and resulting in a decrease the electrical resistance of CaFe_2_O_4_.

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
