# Peer review of "Visible Light-Driven p-Type Semiconductor Gas Sensors Based on CaFe2O4 Nanoparticles"

_sensors, 2020, doi:10.3390/s20030850_

Round 1

Reviewer 1 Report

The works of the authors are meaningful and the paper is interesting. There are several questions/suggestions as follows:

From the SEM image in Fig. 2(b), it seems that the consistency of the sensors even produced at the same batche should not be good due to the non-uniformity of the coating. Have the authors checked this? It will be better to give some explanation about and next possible way to solve this problem. Why did not let the sensor recovered to its original state each time the ethanol was removed in Fig. 4? It’s very important whether the resistance of the senosr may return to the status before the ethanol was entered. Because this is related to the repeatability of a sensor. From Fig. 4 (f), it is found that the sensor’s sensitivity in the dark case is better than other illuminations except the blue one. This is contradictory with the main idea of the paper. Some clerical errors should be corrected. e.g., “According the…” at line 166; “as follows equation” at line 236; “cause an increasing the…”, “resulting a decrease the…” at line 273, 274 etc.

Author Response

Dear Reviewers,

Thank you very much for your comment on our manuscript sensors-704097 entitled “Visible Light‐Driven p-type Semiconductor Gas Sensors Based on CaFe2O4 Nanoparticles”. All the comments have been valuable and beneficial for revising and improving the quality of our manuscript. We have carefully read through the comments and made corrections and modifications accordingly. We are convinced that these modifications will meet your expectations and approval. To allow for an efficient follow-up of the modifications, the revised parts are highlighted in the marked copy of the manuscript. In this response letter, we also use different colours to differentiate between reviewer comments, author response, and change/revision made in manuscript:

Reviewer comment:

Black

Author response or answer:

Green

Change or revision made in the manuscript:

Red

Text from the manuscript

Blue

We would also like to thank you for allowing us to submit a revised version of the manuscript. Should any additional questions or concerns regarding the revised manuscript arise, please feel free to contact me at once. Thank you very much for your consideration.

Qomaruddin

Departament d'Enginyeria Electrònica i Biomèdica
Facultat de Física
Universitat de Barcelona
c/ Martí i Franquès, 1, Pl. 1
08028 Barcelona, Spain

Reviewer 2 Report

The authors present a gas sensor study by monitoring the resistance behavior of calcium iron oxide nanoparticles, forming a porous layer, used as active material deposited over IDEs. The novelty of the work is the investigation of the sensor's behavior at room temperature stimulated by light at different wavelengths. 

The paper presents a well organized study with enough test experiments performed to better understand the sensing mechanism. These experiments include not only dynamic measurements of ethanol detection under illumination but also of other gases (ammonia, nitrogen dioxide) to confirm the validity of the proposed hole concentration change. The above measurements are complemented with control experiments showing the influence of light wavelength on sample resistance without the presence of a gas or under the presence of gas but in dark. All these experiments support the model proposed based on the width modulation of a hole accumulation layer on the surface in a consistent manner.

In terms of sensor performance there are issues related 1) with the continuous increase of the resistance when the gas switches from the on to the off state after each cycle and 2) with the sensor slow time response, however I consider that the experiments and the insight of the paper is good enough to justify its publication.

Some minor remarks.

- In fig. 4d the time is in sec while in all others (fig. 4) in min. Is that correct ?

- In fig. S1 the response of yellow light seems not to follow the general trend. This should be commented in the text.

- What is the thickness of the metal oxide layer ? Is that repeatable from sample to sample ?

Author Response

(The authors gave the same response as above.)

Reviewer 3 Report

See attached file

Author Response

(The authors gave the same response as above.)
